# Circulating miRNAs as Novel Clinical Biomarkers in Temporal Lobe Epilepsy

**DOI:** 10.3390/ncrna10020018

**Published:** 2024-03-17

**Authors:** Lorenza Guarnieri, Nicola Amodio, Francesca Bosco, Sara Carpi, Martina Tallarico, Luca Gallelli, Vincenzo Rania, Rita Citraro, Antonio Leo, Giovambattista De Sarro

**Affiliations:** 1Section of Pharmacology, Science of Health Department, School of Medicine, University “Magna Graecia” of Catanzaro, 88100 Catanzaro, Italy; lorenza.guarnieri@unicz.it (L.G.); francesca.bosco@unicz.it (F.B.); sara.carpi@unicz.it (S.C.); m.tallarico@unicz.it (M.T.); gallelli@unicz.it (L.G.); raniavincenzo1@gmail.com (V.R.); aleo@unicz.it (A.L.); desarro@unicz.it (G.D.S.); 2Department of Experimental and Clinical Medicine, University “Magna Graecia” of Catanzaro, 88100 Catanzaro, Italy; 3Research Center FAS@UMG, Department of Health Science, University “Magna Graecia” of Catanzaro, 88100 Catanzaro, Italy

**Keywords:** drug resistant epilepsy (DRE), antiseizure medications (ASMs), circulating microRNA (c-miRNA), biomarkers, temporal lobe epilepsy (TLE)

## Abstract

Temporal lobe epilepsy (TLE) represents the most common form of refractory focal epilepsy. The identification of innovative clinical biomarkers capable of categorizing patients with TLE, allowing for improved treatment and outcomes, still represents an unmet need. Circulating microRNAs (c-miRNAs) are short non-coding RNAs detectable in body fluids, which play crucial roles in the regulation of gene expression. Their characteristics, including extracellular stability, detectability through non-invasive methods, and responsiveness to pathological changes and/or therapeutic interventions, make them promising candidate biomarkers in various disease settings. Recent research has investigated c-miRNAs in various bodily fluids, including serum, plasma, and cerebrospinal fluid, of TLE patients. Despite some discrepancies in methodologies, cohort composition, and normalization strategies, a common dysregulated signature of c-miRNAs has emerged across different studies, providing the basis for using c-miRNAs as novel biomarkers for TLE patient management.

## 1. Introduction

Epilepsy is a chronic non-communicable neurological disorder characterized by recurrent and spontaneous epileptic seizures. Epilepsy affects more than 51 million individuals across the globe and every year more than 4.9 million people develop new-onset epilepsy, establishing its status as one of the prevalent neurological disorders on a global scale [1]. Epilepsy is marked by recurring seizures, a higher mortality rate, and reduced engagement in social activities and overall quality of life. The basic treatment of epilepsy is pharmacological therapy with antiseizure medications (ASMs) that raise seizure threshold, and thus reduce the propensity for seizures to occur. Nonetheless, about one-third of people with epilepsy do not achieve seizure control and do not respond effectively to current ASMs (drug-resistant epilepsy) and there are no available biomarkers to predict how an individual will respond to particular ASMs [2]. The epilepsy arises from a complex interplay of many genetic and acquired factors, such as altered neuronal excitability, genetic mutations, synaptic abnormalities, imbalances in neurotransmitters, disrupted network connectivity, and inflammatory responses [3]. For most patients, however, the identity of precipitating factors is unknown. The diversity of probable mechanisms and the different possible components of epileptogenesis and ictogenesis suggest various possible biomarker targets [2]. Temporal lobe epilepsy (TLE) is the most common form of refractory focal epilepsy, with an incidence of 40% among epileptic patients. Analysis of brain tissue from refractory epileptic patients and from animal models of TLE shows multi-pathway cellular changes, including neuron loss and gliosis within the hippocampus, inflammation, and reorganization of connections and extracellular structures, and the appearance of recurrent, spontaneous seizures (SRSs) which are resistant to several ASMs [4]. However, the exact underlying mechanisms of epileptogenesis in TLE are still unclear and the development of new ASMs did not provide disease modification in epilepsy or counteract drug resistance in epilepsy. Understanding the pathophysiology of TLE is crucial for the development of supplementary tools needed for disease-modifying or anti-epileptogenic therapies with the primary goal of enhancing patients’ quality of life [5]. The microRNAs (miRNAs), short non-coding RNAs that post-transcriptionally regulate the gene expression landscape, are among potential therapeutic targets and possible biomarkers of pathological changes in the epileptic brain [6]. Abnormal expression of miRNAs is related to various pathophysiological processes, including neuronal death, neuroinflammation, glial cell dysfunction, cell proliferation and migration, neurodegeneration, synaptic remodeling, and neuronal excitability, thus being strongly associated with epileptogenesis [7,8]. Different studies have found extensive dysregulation of various microRNAs (miRNAs) in experimental TLE and human epilepsy, providing evidence for their importance in the pathogenesis of epilepsy, and making them extremely interesting as targets in epilepsy [9]. Furthermore, these studies show that manipulating specific miRNAs can affect seizure thresholds in experimental models of epilepsy [10]. Therefore, dysregulated miRNA profiles in biofluids may be potentially useful biomarkers in people with drug-resistant TLE [11,12]. Regarding circulating microRNAs (c-miRNAs), altered levels have also been found in epilepsy patients [13,14] and in animal models of epilepsy, especially in TLE [15,16]. Through this literature review, the authors aim to provide a comprehensive synthesis of the current understanding of dysregulated c-miRNAs in TLE, with a focus on their potential capability of providing disease modification and counteracting drug resistance in TLE.

## 2. Temporal Lobe Epilepsy: Pathophysiology and Current Understanding

TLE is characterized by SRSs originating in the brain’s temporal lobe, such as the hippocampus or amygdala, and subsequently generalize to other structures. This condition is often, but not always, acquired after a trigger event or injury (infection, traumatic brain injury, stroke, cerebral tumors, febrile seizures) that induces a seizure or status epilepticus (SE). Individuals with TLE frequently encounter a diverse array of cognitive, psychiatric, and behavioral challenges as comorbid conditions along with seizures [17,18]. The International League Against Epilepsy describes two main types of TLE: mesial temporal lobe epilepsy (mTLE), involving the medial or internal structures of the temporal lobe such as hippocampal, parahippocampal gyrus, amygdala, and lateral or neocortical temporal lobe epilepsy (nTLE), a less frequent type, where the onset of seizures is localized in the temporal neocortex [19,20]. mTLE is one of the most common and well-defined forms of symptomatic refractory epilepsy and it is frequently associated with hippocampal sclerosis (HS) characterized by a loss of hippocampal neurons associated with astrogliosis or microgliosis, suggestive of Central Nervous System (CNS) injury and inflammation contributing to epileptogenesis [21,22]. These complex processes result in atrophy and long-term consequences, including cognitive decline and intractability in patients with mTLE [23]. Indeed, about 30% of TLE patients develop pharmacoresistance against available ASMs [24]. For these patients, the only potential relief from recurrent seizures comes from surgical resection of the temporal lobe, including the hippocampus [25]. The diagnosis of epilepsy is complex and subjected to many variations in the patients. Following the preliminary examination with electroencephalography (EEG), structural and functional neuroimaging more elaborative techniques such as Magnetic Resonance Imaging, high-resolution MRI, Positron Emission Tomography, Single Photon Emission Tomography, magnetic resonance spectroscopy, magnetoencephalography, diffusion tensor imaging, and Computed Tomography are available for determinate epileptogenesis foci. The basis of these techniques is different, but they may offer high-quality images that considerably facilitate the diagnosis of TLE. Although these techniques are very helpful in detecting brain abnormal electrical discharges or identifying probable causes of epilepsy, they might lead to false positive or false negative results [26,27]. Poor diagnosis often results in inappropriate pharmacotherapy leading to high morbidity of epileptic patients. Rapid and accurate diagnosis involving the identification of molecular biomarkers easily measurable in biofluids is particularly important in the group of TLE patients. Recently, it has started to emerge the notion that the mechanisms underlying the epileptogenesis, as well as the maintenance and progression of the SE in TLE, involve dysregulated miRNAs that control multiple genes on a system level [28,29]. Therefore, the identification of miRNAs, especially in easily accessible biofluids like blood, sputum or others, has the potential to improve the diagnosis of TLE [30,31]. c-miRNAs create a stable complex with blood proteins or get encapsulated in extracellular vesicles, remaining in circulation for some time after their release. As discussed in this review, c-miRNAs have all the above characteristics to be considered novel potential TLE biomarkers helping the diagnosis, evaluating the risk of developing epilepsy, monitoring, and refractory ASMs therapy.

## 3. microRNAs: Biogenesis and Function

miRNAs are a highly conserved class of short non-coding RNAs that function as negative regulators of gene expression [32]. The biogenesis of miRNAs begins with the transcription of miRNA precursors (pri-miRNA) of several hundred nucleotides (nts) by RNA polymerase II within the cell nucleus. These pri-miRNAs, featuring a stem-loop structure, are initially recognized by the nuclear ribonuclease Drosha, a member of the RNase III family, along with double-stranded RNA-binding protein partners, DiGeorge Syndrome Critical Region 8. Subsequently, the pri-miRNA is cleaved by Drosha, an RNA endonuclease, to generate pre-miRNA, 70 nts with a stem-loop structure, then transported to the cytoplasm through the action of Exportin-5. Here, pre-miRNAs are cleaved to form a double-stranded miRNA molecule consisting of driver and passenger strands, collectively forming the miRNA–miRNA* duplex, with each strand being 20–21 nts in length. This secondary processing is executed by the ribonuclease DICER, an endoribonuclease belonging to the RNase III family, in conjunction with the transactivation-responsive RNA-binding protein (TRBP) [33]. Both strands have the potential to function as mature miRNAs, yet typically, only one, referred to as the guide strand, is integrated into an miRNA-Induced Silencing Complex (miRISC) and acts as a functional miRNA. Conversely, the passenger strand (miRNA*) is typically subject to rapid degradation [33]. miRNAs function through various mechanisms in the regulation of gene expression. The main function is indeed represented by the post-transcriptional mRNA repression, since the miRISC complex brings mature miRNA to the 3′ untranslated regions of specific target mRNAs, leading to the suppression of protein expression and the facilitation of target mRNA degradation. Recent studies have indicated that miRNAs can also interact with ribonucleoproteins in an RISC-independent manner, interfering with their RNA binding functions, a phenomenon known as decoy activity [34]. Interestingly, certain studies have provided evidence that miRNAs can also directly impact gene transcription by binding to the promoter or regulatory regions of DNA. Currently, more than 2000 human precursor miRNAs and nearly 2700 mature miRNAs have been identified in the human genome [35]. However, it is important to note that research in this field is growing, and new miRNAs continue to be discovered. Each miRNA has the potential to influence the expression of hundreds of target genes. Accordingly, the regulation of gene expression by miRNAs has been associated with a plethora of essential biological processes, including regulating neurogenesis, neural differentiation, neuronal excitability, inflammation, apoptosis, and cell metabolism [36]. These processes are frequently altered during the pathophysiological development of various neurological diseases, including epilepsy as TLE [9,37].

## 4. Circulating microRNA-Based Biomarkers in Human Temporal Lobe Epilepsy

c-microRNAs were identified for the first time in 2008 [38] in plasma and serum samples, and subsequently in other body fluids including saliva, breast milk, cerebrospinal fluid, urine, and others [39]. c-miRNAs are packaged in extracellular vesicles (exosomes, microvesicles, apoptotic bodies), or complexed with either RNA-binding proteins (Argonaute) or lipoprotein (high-density lipoprotein). These complexes prevent their degradation from endogenous RNase activity, ensuring their stability in the extracellular environment and enabling their detection at low levels using rapid and dependable techniques like microarray, qRT-PCR or sequencing [40]. c-miRNAs also originate from controlled release, through active secretion, even in apoptotic bodies during physiological cell turnover, but also as a consequence of tissue injury. Moreover, it is known that under pathological conditions, miRNAs have the capability to pass from the brain tissue into the bloodstream through the blood-brain barrier (BBB), making them potentially useful as biomarkers for CNS diseases, such as epilepsy [41,42,43]. In addition, c-miRNAs have been shown as well conserved and involved in neural processes and functions including synaptic transmission, making them extremely interesting as potential drug targets for TLE [16]. Recently, it has been confirmed that a fraction of the circulating pool of miRNAs (miR-19b-3p) in the plasma of mice with kainate (KA)-induced TLE originates from the brain after SE [44]. This finding underscores the potential of brain-derived miRNAs as valuable circulating biomarkers for epilepsy diagnosis. Multiple studies have documented altered expression of c-miRNAs in blood during all phases of the epileptogenesis process, both in animal models of SE and in patients with TLE, providing evidence that epilepsy is associated with extensive changes in miRNA expression [44,45]. The pioneering evidence supporting the notion that brain injuries yield distinctive miRNA profiles in biofluids originates from Liu et al. who demonstrated that KA-induced SE in rats induced distinct alterations in a set of miRNAs within whole blood [46]. Based on this observation, numerous studies have been undertaken, encompassing both experimental models and patient samples, with the greatest part being conducted in the human population from 2015 to 2023 (Table 1).

In 2015, the first investigations on c-miRNAs within the human population were carried out [47,48]. The first study (revealed several serum miRNAs, miR-194-5p, miR-301a-3p, miR-30b-5p, miR-342-5p, and miR-4446-3p), which showed significant differences in drug-resistant patients compared to drug-responsive patients and healthy controls. This study included a total of 303 participants of which, 30 patients with drug-resistant epilepsy, 30 well-response patients, and finally a further 77 drug-resistant patients, 81 drug-responsive patients, and 85 healthy controls in the validation phase. In the discovery and training phase, all serum samples from both the 30 drug-resistant and 30 therapy-responsive patients were sequenced using Illumina HiSeq 2000 technology to select the miRNAs whose expression was altered in the resistant patients compared to the responsive ones. Subsequently, in this study by quantitative gene analysis (qRT-PCR), the serum miRNAs included as the drug-resistant signature were selected. This study for the first time showed that the expression of miR-194-5p, miR-301a-3p, miR-30b-5p, miR-342-5p, and miR-4446-3p in the serum samples of drug-resistant patients was different from that observed in drug-responsive patients, in fact their expression was significantly reduced in drug-resistant patients compared to healthy controls and drug-responsive patients [48].

In 2016, another study showed an upregulation of other c-miRNAs such as miR-143-3p, miR-145-3p, miR-365a-3p, and miR-532-5p in the serum of 15 patients with mTLE-HS, using RT-PCR followed by in-depth bioinformatic analysis of expression levels, 30 min post-seizure [49,50,51]. An increase in miR-129-2-3p plasma levels was documented in samples from patients with refractory TLE compared to healthy volunteers. This three-phase study examined cortical brain tissue samples from nine patients with refractory TLE and eight healthy controls screened for differential miRNA expression. Subsequently, quantitative RT-PCR analysis was performed to evaluate microarrays in brain tissue from a larger number of patients (13 patients with refractory TLE and 13 healthy controls). To further confirm the results, the expression levels by RT-PCR of selected miRNAs in 25 patients with TLE and 25 healthy controls were evaluated in plasma. Microarray analysis showed that miR-129-2-3p expression in both brain tissue and plasma was upregulated in refractory TLE patients compared to controls revealing a significant correlation between this miRNA and seizure frequency. These results suggest that plasma miRNA-129-2-3p could serve as a diagnostic biomarker for detection of refractory TLE [50]. Other authors collected serum during and after the seizure funding an overexpression of miR-378, miR-30a, miR-106b, and miR-15a at seizure onset, compared with levels post-seizure in the serum of patients with epilepsy. When the patients were at seizure onset, the expression of miR-30a was positively associated with seizure frequency suggesting that this miR may be useful for prognostic prediction in epilepsy [52]. Surges et al. collected blood at five different time points from patients with mTLE and compared changes in miRNA expression before and within 30 min, 3–6 h, 20–28 h, and 3–6 days after bilateral convulsive seizures. Additionally, 215 miRNAs were significantly increased within 30 min after bilateral convulsive seizures compared to baseline levels. Out of this miRNA pool, four miRNAs (miR-143-3p, miR-145-3p, miR-365a-3p, miR-532-5p) were successfully validated. The relative abundance of two miRNAs (miR-143-3p and miR-145-5p) in the blood, was inversely correlated with the total seizure duration. It is possible to hypothesize that seizure activity might trigger transient expression of miRNAs that decreases with increasing seizure duration [49].

Subsequently in 2017, three additional case-control studies were published, each contributing valuable insights to the field [53,54,55]. The first evaluation of miR-19b-3p, miR-21-5p, and miR-451a in plasma of TLE and SE patients was correlated with levels of the same miRNAs in cerebrospinal fluid, showing promise in discriminating TLE samples from controls and patients with other neurological diseases or SE [54]. Yan et al. have found that the miRNAs profile of plasma exosomal in patients mTLE-HS was different than healthy controls. Using qRT-PCR analysis, six miRNAs determined in 40 patients with mTLE-HS and 40 healthy controls were selected, and the results showed that exosomal miR-4668-5p, miR-4322, miR-8071, miR-6781-5p, and miR-197-5p were found to be significantly decreased in patients with mTLE-HS, while the expression level of miR-3613-5p was significantly increased [55].

In 2018, there was a nearly two-fold increase in the number of articles published on c-miRNAs in TLE compared to the previous years [56,57,58,59,60]. A set of three blood miRNAs (miR-27a-3p, miR-328-3p, and miR-654-3p) with high diagnostic potential, endowed with the capacity to discriminate between TLE, generalized epilepsy, and control samples have been identified. Bioinformatics analysis of the targets of the three miRNAs identified pathways linked to epilepsy, such as ATP-binding cassette drug transporters (ABCG2 and ABCA1), glutamate transporter SLC7A11 and TP53, and enriched pathways related to growth factor signaling. Furthermore, when examining the combined levels of these three miRNAs in exosomes, diagnostic accuracy increased [59]. The expression of miR-155, by RT-PCR, was found to be significantly higher in the blood serum of patients with TLE (age 72 ± 6.5) than in the healthy controls (age 68 ± 5.5), thus proving that miR-155 contributed to epileptogenesis and exhibited a neuroprotective effect on epilepsy-induced neuronal apoptosis via the PI3K/Akt/mTOR signaling pathway [57].

From 2019 to 2023, other studies investigating c-miRNAs in patients with TLE were published. The over-expression of miR-145, miR-181c, miR-199a, and miR-1183 in the blood of patients with drug-resistant mTLE-HS was identified [61]. Hippocampal and blood samples were collected from 20 patients who underwent amygdalohippocampectomy due to pharmacoresistant mTLE-HS (10 favorable surgical outcomes, Engel I and 10 with unfavorable surgical outcomes, Engel III–IV), and from 10 healthy controls. RT-PCR was used to analyze the expression of miR-145, miR-181c, miR-199a, and miR-1183, showing that the expressions of these miRNAs differed quantitatively in the hippocampus and blood of the mTLE-HS patients compared to the respective control. The miR-145, miR-181c, miR-199a, and miR-1183 were overexpressed in the blood of mTLE-HS patients, and this difference was more pronounced for miR-145, with no significant difference in the blood and hippocampus of Engel I patients and Engel III–IV patients [61]. Moreover, it was reported the downregulation of miR-145-5p in the plasma of patients with drug-resistant mTLE compared to healthy controls. This study cohort enrolled 40 patients with refractory epilepsy, including 11 with mTLE and 42 controls, and several plasma samples were collected from each patient for analysis by RT-PCR of miR-145-5p expression levels. The significant downregulation of plasma miR-145-5p in patients with refractory epilepsy and mTLE compared to healthy controls was reported, suggesting that its lower expression was associated with earlier age at epilepsy onset and more frequent seizures [62]. Overall, both studies identified significant alterations of c-miRNAs. It remains, however, unclear whether these changes in serum miRNA profiles reflect chronic changes associated with the underlying disease. In children’s patients, miR-146a and miR-106b plasma expression was significantly upregulated than that in normal control, suggesting that these upregulated plasma microRNAs could be used as biomarkers for epilepsy evaluation [63]. Another study by Zhao et al. has confirmed that miR-106b was more highly expressed in children with TLE than in the healthy group [64].

In 2020, a study found lower expression levels of miR-139-5p in the serum samples of children with refractory epilepsy compared with the serum from newly diagnosed epilepsy children and normal children. This decreased expression of miR-139-5p was accompanied by robust expression of Multidrug-resistance protein 1 (MRP1). Indeed, miR-139-5p exerts a negative regulation on MRP1 expression. Similarly, in refractory epilepsy, drug sensitivity was improved as a result of the upregulation of miR-139-5p through the downregulation of MRP1 that reduced apoptosis and promoted survival of neurons alleviating neuronal damage. Therefore, the upregulation of miR-139-5p could be a promising tool for the prevention of epilepsy progression and the reduction in ASMs resistance [65].

Other studies have identified a set of dysregulated blood plasma microRNAs comprising increased miR-93-5p, miR-142-5p, miR-182-5p, miR-199a-3p and decreased miR-574-3p during epileptogenesis or chronic epilepsy in rodent models of TLE. Validation studies found miR-93-5p, miR-199a-3p, and miR-574-3p were also dysregulated in plasma from patients with intractable TLE. The dysregulation of these specific miRNAs seems to be related to the development of disease rather than seizure [66]. Regarding miR-15a, significant downregulation in the serum of children with TLE has been found, suggesting its potential role in the pathogenesis of this disease. A significant relationship between this miRNA with TLE-induced reduction in cell viability and TLE-induced cell apoptosis has also been demonstrated. Nonetheless, miRNA-15a-5p seems to have great potential to be used as a biomarker for the diagnosis of TLE in children, since it showed high sensitivity and specificity [67]. Another study has found a significant dysregulation of miR-194-5p in the plasma of children with TLE than in healthy children, suggesting a high specificity and sensitivity of this miR-194-5p in the diagnosis of TLE in children [68].

In addition, RT-PCR analysis of three serum miRNAs, specifically miR-142-5p, miR-146a-5p, and miR-223-3p, were found to be significantly upregulated in 27 patients with TLE (17 drug responsive and 10 drug-resistant) compared to 20 control subjects. Moreover, results found from a logistic regression model demonstrated the effectiveness of miR-142-5p and miR-223-3p in distinguishing drug-sensitive to drug-resistant TLE patients [69]. Subsequently, by analyzing the expression profiles of miR-629-3p, miR-1202, and miR-1225-5p, in the blood of 20 patients with mTLE-HS (10 with favorable surgical prognosis Engel I, and 10 with unfavorable surgical prognosis, Engel III-IV) and 10 control subjects, it was shown that the expression of these miRNAs was upregulated in the blood of mTLE-HS patients compared to the control group [70]. The most recent research in the field of c-miRNAs and mTLE has reported a reduction in miR-22 expression levels in the serum of 40 mTLE-HS patients (10 drug-responsive and 30 drug-resistant) compared to 48 control blood donors. The down-modulation of the miR-22 in serum was inversely related to P2X7R overexpression in the hippocampus and temporal neocortex of human mTLE-HS patients [71] as also reported in the experimental TLE animal model [72]. Upregulation of the P2X7R in the brain may affect pathological processes associated with the severity and propagation of seizure. The inverse relationship between miR-22 and P2X7R implies the measure of serum miR-22 as a possible clinical surrogate of P2X7R brain expression in drug-resistant mTLE-HS patients [71].

In a study conducted in 2023, the serum levels of miR-146a-5p and miR-132-3p were measured in 40 adult epileptic patients and 40 control samples. RT-PCR analysis has found that serum expression levels of miR-146a-5p and miR-132-3p were significantly higher in epileptic patients compared to controls. In addition, a significant difference was observed in the expression of miRNA-146a-5p in the focal group when non-responder patients were compared to responder groups. Further analysis showed that increased seizure frequency was the only risk factor that could influence the drug’s response. Thus, miR-146a-5p and miR-132-3p may be involved in epileptogenesis and together these c-miRNAs may be useful as a diagnostic biomarker, but they are not predictors of ASMs response [73].

From all these studies, it emerges that several c-miRNAs are dysregulated in epileptic brains, and such dysregulation is involved in cellular processes correlated to the mTLE both in adult and children patients [74,75,76]. In the next section, we will provide a detailed analysis of six c-miRNAs as potential biomarkers. These miRNAs have been chosen based on their significant dysregulation in at least two separate studies conducted on biofluids from TLE patients.

**Table 1 ncrna-10-00018-t001:** Circulating microRNA as biomarker in patients with TLE.

Significantly Dysregulated xc-miRNAs	Source	Cohort Composition	Technique	References
↓ miR-30b-5p (drug-resistant vs. drug-responsive, drug-resistant vs. Ctrl) ↓ miR-194-5p (drug-resistant vs. drug-responsive, drug-resistant vs. Ctrl) ↓ miR-301a-3p (drug-resistant vs. drug-responsive, drug-resistant vs. Ctrl) ↓ miR-342-5p (drug-resistant vs. drug-responsive, drug-resistant vs. Ctrl) ↓ miR-4446-3p (drug-resistant vs. drug-responsive, drug-resistant vs. Ctrl)	Serum	Discovery and training set: 30 TLE drug-resistant 30 TLE drug-responsive Validation set: 77 TLE drug-resistant 81 TLE drug-responsive 85 controls	Discovery set: Illumina HiSeq 2000 technology Training and validation set: RT-PCR	[16]
↑ miR-134 (TLE vs. Ctrl)	Plasma	2 healthy controls 5 TLE drug-resistant 1 JME	RT-PCR	[47]
↑ miR-129-2-3p (TLE vs. Ctrl)	Plasma	25 TLE 25 healthy controls	RT-PCR	[50]
↓ miR-153-3p (mTLE vs. Ctrl)	Plasma	32 surgical patients with mTLE 18 surgical controls 56 mTLE 101 healthy non-surgical controls	RT-PCR	[51]
↑ miR-143-3p (30 min post-seizure vs. pre-seizure) ↑ miR-145-3p (30 min post-seizure vs. pre-seizure) ↑ miR-365a-3p (30 min post-seizure vs. pre-seizure) ↑ miR-532-5p (30 min post-seizure vs. pre-seizure)	Serum	Validation set: 15 mTLE-HS (pre-seizure and 30 min following a single focal seizure evolving to a bilateral convulsive seizure)	RT-PCR	[49]
↓ miR-134 (mTLE vs. Ctrl; mTLE drug-responsive vs. Ctrl; mTLE drug-resistant vs. Ctrl)	Plasma	Discovery set: 14 mTLE 13 focal cortical dysplasia 16 healthy controls Validation set: 65 mTLE (27 drug-responsive and 38 drug-resistant) 83 healthy controls	RT-PCR	[53]
↓ miR-4668-5p (TLE-HS vs. Ctrl) ↓ miR-4322 (TLE-HS vs. Ctrl) ↓ miR-8071 (TLE-HS vs. Ctrl) ↓ miR-6781-5p (TLE-HS vs. Ctrl) ↓ miR-197-5p (TLE-HS vs. Ctrl) ↑ miR-3613-5p (TLE-HS vs. Ctrl)	Plasma exosomal	Validation set: 40 mTLE-HS 40 healthy controls	RT-PCR	[55]
↓ miR-153-3p (mTLE vs. Ctrl)	Plasma	22 surgical patients with mTLE 20 controls (head trauma or cerebral hemorrhage)	qRT-PCR	[58]
↑ miR-27a-3p (EAS vs. Ctrl; EAS vs. EBS; TLE vs. Ctrl; GGE vs. Ctrl) ↑ miR-328-3p (EAS vs. Ctrl; EAS vs. EBS; TLE vs. Ctrl) ↓ miR-328-3p (EBS vs. Ctrl) ↑ Ago2-miR-328-3p (EAS vs. Ctrl; EAS vs. EBS) ↓ miR-654-3p (EBS vs. Ctrl; EBS vs. EAS) ↑ miR-654-3p (GGE vs. Ctrl)	Plasma (Complexed to Argonaute2 or bound in exosomes)	Discovery set: 32 TLE, samples was collected: EAS, EBS 32 healthy controls Validation set: 102 TLE 10 FLE 23 GGE 6 SE 15 PNES 110 healthy controls	RT-PCR Digital PCR	[59]
↑ miR-155-5p (TLE vs. Ctrl)	Serum	TLE healthy controls	RT-PCR	[57]
↑ miR-301a-3p (mTLE vs. Ctrl)	Plasma and a sample of tissue from the left hippocampal region	1 mTLE drug-resistant (Sudden and unexpected death in epilepsy) 10 controls (traumatic or asphyxia deaths)	TaqMan assays	[56]
↑ miR-145 (Engel I vs. Ctrl) ↑miR-181c ↑ miR-199a (Engel I vs. Ctrl) ↑ miR-1183 (Engel I vs. Ctrl; Engel III-IV vs. Ctrl)	Blood	20 patients who underwent amygdalohippocampectomy due to pharmacoresistant mTLE-HS (10 Engel I and 10 Engel III–IV) 10 healthy controls	RT-PCR	[61]
↓ miR-145-5p (Refractory epilepsy vs. Ctrl; mTLE vs. Ctrl)	Plasma	40 patients with refractory epilepsy (11 mTLE) 42 healthy controls	RT-PCR	[62]
↑ miR-328-3p (mTLE-HS vs. Ctrl; Engle I vs. Ctrl; Engle III-IV vs. Ctrl) ↑ miR-654-3p (Engle I vs. Ctrl)	Serum	28 mTLE-HS (14 Engle I, 14 Engel III–IV) 11 healthy controls	RT-PCR	[45]
↑ miR-142-5p (TLE vs. Ctrl; TLE drug-resistant vs. TLE drug-responsive) ↑ miR-146a-5p (TLE vs. Ctrl) ↑ miR-223-3p (TLE vs. Ctrl; TLE drug-resistant vs. TLE drug-responsive)	Serum	27 TLE (17 drug-responsive; 10 drug-resistant) 20 healthy controls	RT-PCR	[69]
↑ miR-629-3p (mTLE-HS vs. Ctrl; Engel I vs. Ctrl; Engel III-IV vs. Ctrl) ↑ miR-1202 (mTLE-HS vs. Ctrl; Engel I vs. Ctrl) ↑ miR-1225-5p (mTLE-HS vs. Ctrl; Engel I vs. Ctrl; Engel III-IV vs. Ctrl)	Blood	20 mTLE-HS (10 Engel I; 10 Engel III–IV) 10 healthy controls	RT-PCR	[70]
↓ miR-22 (mTLE-HS vs. Ctrl; drug-resistant vs. Ctrl)	Serum	40 mTLE-HS (10 drug-responsive; 30 drug-resistant) 48 healthy controls	RT-PCR	[71]

(↓ = reduced expression; ↑ = increased expression).

## 5. Candidate c-miRNAs with Biomarker Potential in TLE Patients

### 5.1. miR-27a-3p

miR-27a-3p is associated with inter-endothelial junction loss and barrier integrity disruption [77]. In the context of TLE, a systematic review and meta-analysis identified miR-27a-3p as one of the most consistently upregulated miRNAs across an analyzed set of expression profiles (three human TLE-HS studies and 11 experimental animal models of post-SE). Specifically, miR-27a-3p is among the eight consistently upregulated mature miRNAs in the latent stage of TLE [78]. Notably, in experimental kainate-induced TLE, the upregulation of miR-27a-3p produced neuronal death in the hippocampus after SE. In contrast, the inhibition of miR-27a-3p prevented epilepsy-induced inflammatory responses and hippocampal neuronal apoptosis by targeting mitogen-activated protein kinase 4 (MAP2K4) in the same model of TLE [79].

Moreover, in a litio–pilocarpine-induced TLE model, miR-27a-3p was able to regulate multiple ion channel-related differentially expressed genes as KCNB1, SCN1B, and KCNQ2 in mTLE [80]. Regarding circulating levels of miR-27a-3p, it has shown the potential to discriminate TLE patients from healthy controls, and it represents potential molecular seizure biomarkers, distinguishing post-seizure samples from baseline samples in plasma and exosomes. A 2018 study analyzed the plasma of 32 patients with TLE and 32 healthy patients, and the plasma of patients with TLE was divided into epilepsy baseline samples (before seizures, EBS) and after-seizure samples (after seizures, EAS). For validation, the expression levels of miR-27a-3p, miR-328-3p, and miR-654-3p were assessed in 102 patients with TLE (29 EBS and EAS), in 110 controls, 10 patients with frontal lobe epilepsy (FLE), 23 with genetic generalized epilepsy (GGE), 6 patients with SE, and finally 15 patients with seizures (PNES). In the first analysis, the authors observed that the miRNAs miR-27a-3p, miR-328-3p, and miR-654-3p were highly specific for patients with FLE compared to healthy patients and patients with GGE, whereas no significant differences were observed between healthy controls and patients with PNES. RT-PCR and digital PCR analysis especially in the exosome-enriched fraction showed good diagnostic accuracy, and it was also seen that Argonauta-complexed miR-328-3p was increased together with miR-27a-3p in patients with seizures, in contrast to miR-654-3p which was hypo-expressed in patients with TLE, thus demonstrating that these miRNAs can be considered important diagnostic biomarkers for epilepsy patients [59].

In pediatric patients with epilepsy, miR-27a-3p plasma levels were also lower than in healthy children as reported in adult patients; miR-27a-3p reduced the levels of Limk1, suggesting that Limk1 is a target of miR-27a-3p [81].

Bioinformatics analyses of targets related to miR-27a-3p underscore possible correlations with pathways linked to neuronal excitability and epilepsy [79]. Thus, miR-27a-3p plays an important role in the inflammatory response and apoptosis of hippocampal neurons in epilepsy.

### 5.2. miR-134

miR-134 has been identified as a key regulator in multiple facets of neuronal function. In the adult brain, it is constitutively expressed in the dendrites and in the neurons bodies [82] and in the CA1 and CA3 hippocampal regions [10]. The role of miR-134 is closely related to protein kinase LIM kinase-1 (Limk-1)-dependent dendritic spine development, synaptic transmission, and synaptic plasticity [83,84]. Also, miR-134 is a critical regulator of homeostatic synaptic repression, modulating synaptic strength by targeting Pumilio-2, underscoring its intricate role in maintaining the balance of synaptic function [85].

In the context of epilepsy, increased brain levels of miR-134-5p have been found in studies on animal models of TLE [76,83,86,87] and in tissues of patients with refractory TLE indicating miR-134 upregulation as a common response to drug-resistant TLE [10,82]. Increased miR-134 was linked by reduced protein levels of both Limk1 and CREB, suggesting these as possible targets in vivo [10]. Moreover, the increased circulating levels of miR-134 were strongly correlated with temporal lobe sclerosis. Injection of miR-134-targeting antagomirs blocked seizure-induced upregulation of miR-134, with consequent decrease in the subsequent occurrence of SRSs in the chronic stage of epileptogenesis and exerting neuroprotective effects in experimental models of TLE. The anti-epileptogenic effect seems to be partially mediated via the rescue of the LimK1 protein, a critical regulator of dendritic spine structure [10,87,88].

Studies of c-miR-134 validation as biomarkers of mTLE, showed that this miR can be the promising diagnostic biomarker of this epileptic disorder, although there is some controversy about the circulating levels of miR-134.

Regarding human epilepsy, studies conducted on c-miR-134 have found that it represents the most promising diagnostic biomarker in mTLE. In a study conducted by Avansini et al., quantitative RT-PCR was used to measure plasma levels of miR-134 in two study phases: an initial discovery phase with 14 patients with mTLE, 13 with focal cortical dysplasia and 16 controls; a validation cohort consisting of an independent cohort of 65 patients with mTLE and 83 controls. From the microRNA analysis, a significant downregulation of miR-134 was observed in the plasma of 14 patients with mTLE with HS and focal cortical dysplasia, compared to 16 healthy controls (*p* < 0.001), regardless of their response to treatment with ASMs or the presence of signs of HS on MRI. Therefore, this indicated that circulating miR-134 might act as a biomarker for mTLE itself rather than as a measure of response to ASM treatment [53]. In contrast, the same miRNA was documented in a small cohort study of patients with TLE. Plasma from the patients enrolled in the study was obtained from two healthy volunteers, five patients with drug-refractory TLE, and one patient with juvenile myoclonic epilepsy (JME was collected with a confirmed seizure-free monitoring period of at least 24 h to perform RNA extraction. Real-time PCR analysis using the TaqMan miRNA assay revealed very low levels of miR-134 in the samples of the healthy volunteers, whereas plasma miRNA levels were consistently higher in the TLE patients [47]. Thus, miR-134 upregulation seems to be a common response to pathologic brain activity, and the level of miR-134 changes in the plasma of TLE patients suggests that miR-134 may have diagnostic power [89]. In a study involving patients with various forms of newly diagnosed epilepsy, increased miR-134 expression was reported in patients with new onset epilepsy and SE, suggesting that the presence of higher plasma miRNA-134 levels at the onset of epilepsy may accompany the progression and severity of epileptic seizure. Therefore, at the onset of epilepsy, the determination of plasma levels of miR-134 may be useful to determine the severity and duration of the epileptic seizure as it reflects the extent of the underlying pathophysiological processes [90]. Patients with increased circulating miR-134 levels are at significantly higher risk for developing drug-resistant epilepsy, compared to those with favorable responses to ASMs, independently of temporal lobe sclerosis and other factors [91]. These recent studies suggest that it might be possible to modify epileptogenesis using strategies that target miR-134. Regarding pediatric epilepsy patients, only one study was conducted, showing upregulation of miR-134 in the plasma from pediatric patients compared with controls [81].

### 5.3. miR-153-3p

miR-153-3p exhibits high expression in brain tissue and is involved in the neuro-behavioral development during the early stage of neurogenesis [92]. Regarding epilepsy, it has been reported as significantly downregulated in the plasma of patients with drug-resistant mTLE. In this study, 56 patients diagnosed with mTLE undergoing treatment (of which 32 patients had undergone anterior temporal lobectomy), 18 patients undergoing surgical treatment for head trauma or brain hemorrhage, and 101 healthy controls of the same age were enrolled. miRNA expression analysis by RT-PCR on the temporal cortex showed a significant downregulation of miR-153-3p in the plasma of 32 mTLE surgical patients compared to 18 surgical controls, and downregulation of miR-153-3p was also validated in 56 mTLE patients with/without surgery compared to 101 healthy non-surgical controls. Notably, aberrant expression of miR-153-3p is associated with the upregulation of hypoxia-inducible factor-1 alpha (HIF-1α) in the temporal cortex of patients with mTLE [51]. The inverse correlation between the levels of HIF-1α and miR-153 underscores the significance of this miRNA in epileptic context. In a study conducted by Gong et al. to investigate the role of miR-153 in mTLE, its expression was assessed in 22 surgical cases of mTLE and 20 controls. Their data found that miR-153 was downregulated in mTLE patients compared to controls, and qRT-PCR showed that HIF-1α expression levels were upregulated in the temporal cortex of mTLE patients. These results showed that there is a negative correlation between miR-153 and HIF-1α expression levels [58].

Nowadays, it is known that HIF-1α actively contributes to the pathogenesis of pharmacoresistant epilepsy. This contribution is, in part, due to the upregulation of P-glycoprotein (P-gp), highlighting a multifaceted interplay between miR-153-3p, HIF-1α, and P-gp in molecular basis of resistance to pharmacological interventions [58]. From both studies, it emerges that miR-153, through the regulation of HIF-1α expression, contributes to the pathogenesis of drug-resistant epilepsy [51,58]. The convergence of these independent findings underscores the biological significance of the observed miR-153 downregulation across diverse patient cohorts, suggesting its potential role as a marker in the context of drug-resistant mesial TLE. All these data demonstrate that miR-153-3p is a key factor in epileptogenic processes, especially in drug-resistant TLE in both pre-clinical and clinical models of epilepsy. In pediatric patients, however, no data on miR-153-3p have been reported, instead, the only data that seems to have had an important diagnostic value in distinguishing pediatric patients with epilepsy from healthy children, were observed with miR-155, which seems to be a new diagnostic biomarker particularly related to both the degree of the disease and its course [93,94]. In this regard, Liu et al. demonstrated that the serum expression levels in exosomes of miR-155 were respectively high in epileptic children, and its expression levels also seemed to correlate not only with the disease course but also with the presence of an abnormal EEG, although the authors did not analyze whether there was a correlation between EEG findings and miR-155 expression [93].

### 5.4. miR-301a-3p

Similar to miR-153-3p, miR-301a-3p exhibits heightened expression in brain tissues compared to other tissues. In the context of epilepsy, miR-301-3p downregulation is documented in the hippocampus in a lithium-pilocarpine-induced SE rat model [95], as well as in the dentate gyrus of rats following amygdala stimulation-induced SE [96]. Similarly, it has been reported a significant downregulation of miR-301-3p in the hippocampus of pharmacoresistant TLE patients compared to controls [97]. In contrast, Kan et al. found an upregulation of miR-301a-3p in hippocampal tissue samples of patients with TLE, both with and without HS, compared to the hippocampus of autopsy control patients [98]. In biofluids, miR-301-3p displayed a significant decrease in the serum of drug-resistant patients compared to drug-responsive patients and healthy controls and it was negatively associated with seizure severity. Thus, miR-301a-3p demonstrated the highest diagnostic value for drug-resistant epilepsy and was inversely linked with seizure severity. The drug-resistant mechanism linked to impaired expression of this miR-301a-3p is correlated with an increased expression of NF-kB protein, supporting a role in inflammation and apoptosis [16,55]. However, controversial data also emerged in biofluids. De Mattes et al. documented, by TaqMan Assay, a marked upregulation of miR-301a-3p in both the hippocampus and plasma of a patient experiencing sudden and unexpected death in epilepsy due to drug-resistant mTLE in comparison to 10 autopsies for traumatic or asphyxia deaths [56]. Moreover, upregulated miR-301a-3p in the blood plasma from pediatric epilepsy patients has been found compared to those from healthy children [81]. The apparent inconsistency across these findings may arise from diverse factors, including technical variations, different standard methods, limited sample size, different study design, individual characteristics, lifestyle factors, among others, highlighting the complexity of interpreting c-miRNAs analyses.

### 5.5. miR-328-3p

Similar to miR-153-3p and miR-301a-3p, miR-328-3p displays elevated expression in brain tissues compared to other tissues. Despite this, research exploring its physiological role in neurological functions is lacking. Several articles reported this miRNA to function as both a tumor suppressor and an onco-miRNA in different tumors [99]. In the context of epilepsy, miR-328 levels significantly decrease in the rat hippocampus 48 h following pilocarpine-induced SE [100]. Regarding its potential as a biomarker, miR-328-3p was initially evaluated in a case-control design study conducted on plasma samples from drug-resistant epilepsy patients from two different centers. They reported that miR-328-3p plasma copy number in samples of TLE patients was significantly lower than that of controls, even in exosome-enriched fraction, effectively discriminating mTLE patients from healthy individuals [59]. In line with the clinical observation, the levels of miR-328-3p in plasma samples from mice with intra-amygdala kainic acid-induced TLE were significantly lower compared to those in the control group. Moreover, when comparing levels in patients with or without HSs, Raoof et al. found no significant differences in plasma levels of miR-328-3p between TLE patients with or without HS [59]. Ioriatt et al. also assessed this miRNA in the serum of patients with mTLE-HS, finding that the increase in the expression levels of miR-328-3p exhibited significant discriminatory power between healthy subjects and epilepsy patients, but no discriminative power in prognosis was identified when comparing patients with mTLE-HS (*n* = 28) with good surgical prognosis (Engel I group, *n* = 14) versus patients with mTLE-HS with unfavorable surgical prognosis (Engel III–IV group, *n* = 14) [45].

All these data on miR-328-3p suggest that there may be a link between levels of this biomarker in the bloodstream and seizure severity, emphasizing the need for further investigation in larger cohorts. Finally, a study on pediatric epilepsy patients found that levels of miR-328-3p were lower in the plasma of epileptic children than in healthy children, but this difference was not statistically significant [81].

### 5.6. miR-654-3p

miR-654-3p is implicated in the different biological pathways such as glutamate transporter SLC7A11, TP53, and ATP-binding cassette drug transporters. GLRA2, the glycine receptor subunit, is a specific target of miR-654-3p [66]. Several research articles pointed to the involvement of miR-654-3p in tumorigenesis across various contexts [101,102]. However, in epilepsy research, there are only two studies regarding miR-654-3p [45,59]. The study of Ioriatti reported significantly elevated levels of miR-654-3p in mTLE-HS patients (*n* = 28) with good surgical prognosis (Engel I, *n* = 14) compared to controls (*n* = 11), as well as Engel I compared to mTLE-HS with unfavorable surgical prognosis (Engel III–IV, *n* = 14), suggesting that miR-654-3p may serve as a potential marker for the surgical prognosis of mTLE-HS [45]. c-miR-654-3p levels demonstrated promising performance in differentiating patients with generalized seizures from individuals with TLE, and a similar pattern of downregulation of c-miR-654-3p has been found in samples from mice with intra-amygdala kainic acid-induced TLE or in plasma samples from mice with intra-amygdala kainic acid-induced TLE [59]. Based on all these data, it seems that c-miR-654-3p could potentially represent a diagnostic biomarker of TLE and a prognostic biomarker of the surgical outcome of epilepsy. In children with new-onset epilepsy, the plasma levels of miR-654-3p are more increased than healthy controls [81].

## 6. Insights into the Effect of miRNAs Target on Antiseizure Medication

Data on the influence of ASM on these six microRNA expression levels in body fluids are not consistent. Some studies have indicated that some ASMs can impact the dysregulated plasmatic expression miRNAs in both experimentally-induced epilepsy and epileptic patients. The effects of diazepam and carbamazepine, two ASMs that are more or less effective at suppressing SRS in mouse models of TLE, have been investigated on dysregulated plasmatic levels of miR-27a-3p, miR-328-3p, and miR-654-3p after intra-amygdala kainic acid-induced SE in mice. Administration of diazepam or carbamazepine did not alter the plasmatic levels of the three dysregulated miRNAs in epileptic mice [59,66]. These data suggest that levels of miRNA are more strongly correlated with underlying pathophysiology and disease processes rather than external influences or acute anti-seizure effects. The same ASM (diazepam or carbamazepine) used in three different models of epileptogenesis did not induce significant changes in plasma levels of other miRNAs (miR-199a-3p and miR-142-5p) reported in this paper. However, it cannot be excluded that other drugs may affect the levels of these miRNAs [66]. For example, silencing of miR-155, another miRNA reported in this review, by administering miR-155 antagomir, improved KA-induced seizures, electroencephalogram, and proinflammatory cytokine expression by inducing a morphological change in microglia. All these data therefore suggest that abnormal overexpression of miR-155 may contribute to epileptogenic processes through the induction of microglial neuroinflammation [103]. Regarding miR-134, its plasma level in the patients with severe epilepsy was significantly decreased after one month of treatment with valproic acid. This indicates that the plasma miRNA-134 level was elevated in epileptic patients, and the valproic acid treatment probably normalized the plasma miRNA-134 levels through downregulation of BDNF, in turn affecting the binding of miR-134 to LIM domain kinase-1 mRNA. Therefore, the elevated miRNA-134 plasma level can be likely used as a biomarker in epileptic seizure, while the decrease in plasma miRNA-134 possibly results from efficient anti-epileptic treatment. miR-134 levels have also been linked with the duration and severity of epileptic seizures before treatment, indicating that miR-134 would be feasible as a response-monitoring biomarker or novel therapy target rather than as a response-predictive marker [90].

Regarding other c-miRNAs reported in this review, c-miR-106b has been found to be more highly expressed in children with TLE than in the normal group, suggesting that miR-106b can be used as a potential diagnostic indicator for children with TLE. In pediatric patients, miR-106b expression levels were reduced following treatment with levetiracetam in combination sodium valproate, thus demonstrating that this miRNA could be considered a potential diagnostic marker for children with epilepsy after treatment [64].

## 7. Diagnostic and/or Prognostic Role of c-miRNAs in TLE

On the basis of above-reported findings, it is likely to use some miRNAs as markers for treatment outcome, while others to monitor disease progression. Therefore, in this context, miR-27a-3p, miR-134-5p, miR-153-3p, miR-301a-3p, miR-328-3p, and miR-654-3p seem to require special attention. The connection of miR-27a-3p, miR-328-3p, and miR-654-3p with epileptogenesis has been demonstrated by previous data, according to which overexpression of miR-27a-3p, miR-328-3p, and hypo-expression of miR-654-3p occur in patients with TLE and can be considered as diagnostic biomarkers for this condition [59].

Furthermore, miR-27a-3p appears to be consistently upregulated in the latent stage of TLE [78]. The overexpression of miR-328-3p also showed significant discriminatory power between healthy subjects and patients with epilepsy, but no discriminatory power in prognosis was identified when comparing patients with mTLE-HS with good surgical prognosis and patients with mTLE-HS with poor surgical prognosis, whereas its overexpression allowed us to discriminate patients with mTLE from healthy controls [59]. Regarding miR-654-3p, it was also shown a high variability of its expression. The connection, on the other hand, of miR-134 with epilepsy has been demonstrated by numerous studies, in which it was reported that its expression can also occur during short-term generalized seizures, thus indicating that under certain pathophysiological conditions, not only it affects neuronal activity but can also directly influence epileptogenic and pathogenic brain activity. It was found to be an important biomarker of chronic phase TLE, and its overexpression can occur following post-convulsive neuronal damage. These results therefore demonstrate that miR-134 may act as a biomarker of TLE itself rather than as a measure of response to ASM treatment [53,76]. The miR-153-3p was shown to be an important biomarker in the context of drug-resistant mTLE as its expression levels were downregulated in these patients, and thus allowed to distinguish patients with drug-resistant mTLE from healthy subjects [58]. Finally, miR-301-3p was also shown to be an important biomarker in the context of drug-resistant epilepsy, as its altered expression correlated with increased neuronal inflammation, and its data could provide the best values for diagnosis in patients with drug-resistant epilepsy [55,56].

Of note, miR-146, plasma miR-134, plasma miR-145-5p, serum miR-301a-3p, and miR-153 have been reported as potential diagnostic biomarkers for drug-resistant epilepsy [16,51,62,91]. These c-miRNAs represent an opportunity as a highly sought-after predictive biomarker in the therapeutic management of these epileptic drug-resistant patients. Potential diagnostic biomarkers for TLE are serum miR-223, serum miR-142 [69], plasma miR-199a, plasma miR-574-3p, blood plasma miR-27a-3p, blood plasma miR-328-3p, and blood plasma miR-654-3p [59,66]. Potential diagnostic biomarkers for mTLE are represented by plasma miR-145-5p [62] and plasma miR-134 [53]. Potential diagnostic biomarkers for mTLE with HS are reported to be blood miR-145, blood miR-181c, blood miR-199a, blood miR-1183 [61], plasma exosomes miR-3613-5p, miR-4668-5p, miR-8071, miR-197-5P [55], serum miR-328-3p, and serum miR-654-3p [45]. Instead, circulating miR-654-3p and miR-145-5p have been proposed as potential prognostic biomarkers of mTLE-HS [45] and mTLE [62], respectively. Prognostic biomarkers for drug-resistant TLE are represented by serum miR-223, serum miR-142 [69], serum miR-146, and serum miR-134 [91].

## 8. Different miRNAs Behavior and Their Variability as Biomarkers

The study by Avansini et al. revealed a statistically significant decrease in plasma miR-134 levels in patients with mTLE compared with the control group; therefore, the authors concluded that such a decrease represents a potential diagnostic biomarker of mTLE [53]. In contrast, a study on five patients with TLE and one patient with JME has found increased expression of circulating miR-134 in the blood of these patients, which led them to conclude that miR-134 may be a useful marker in refractory epilepsy [47]. On the other hand, it has been shown that overexpression of miR-134 occurred in some areas of the hippocampus after seizures following local post-convulsive neuronal damage. This overexpression was observed in a mouse model of mTLE and in surgically obtained hippocampal tissue samples from patients with therapeutically resistant mTLE [10].

Also, in a mouse model of mTLE, animals that overexpressed miR-134 at the level of the hippocampus presented a 50% higher seizure frequency and risk of developing the epileptic state than animals in which miR-134 expression did not change [82]. Based on these and other studies, it might therefore appear that the results of the study by Avansini et al. are at odds with the rest of the evidence to date [53]. It must be said, however, that although the discovery of the regulation of miRNAs in biofluids, including blood, has made the hypothesis of using them as non-invasive diagnostic biomarkers very attractive; however, much remains to be understood about their expression in various brain areas and blood in conjunction with epilepsy, especially considering that the expression of such c-miRNAs can vary so much depending on the form of epilepsy studied. Thus, it is not surprising that miR-654-3p behaves differently in different forms of epilepsy [45,59] or that miR-301-a-3p is hypo-expressed in the hippocampus of patients with pharmacoresistant TLE [16] and conversely overexpressed in the hippocampus and blood of patients with drug-resistant mTLE [56].

Regarding miR-134, although neurons appear to be the main cell type expressing it, there remains some uncertainty, however, about which subtypes of neurons more specifically express it. Two works support the hypothesis of broad expression in both excitatory and inhibitory neurons [84,89]. However, an increased expression of miR-134 in some populations of inhibitory interneurons has also been suggested [89]. Addressing this point would be important because modulation of miR-134 levels could have opposite effects if it is predominantly expressed in inhibitory rather than excitatory neurons.

Even more fascinating and complex then turns out to be the hypothesis that miR-134 can function differently, through distinct molecular mechanisms, depending on the specific brain states at hand [89]. Indeed, not only epilepsy but also other neurological disorders can cause increased plasma levels of miR-134, and this raises the issue that it may not be specific enough as a biomarker of epilepsy [104,105]. Another explanation for the discordance in the data could be provided by the fact that miRNAs are brain-specific, and thus their presence in the circulation reflects possible brain damage or alteration of the BBB that is known to accompany seizures, but they are also susceptible to intense changes in a short time [44]. In addition, it has been reported lower levels of miR-134 in the plasma of mTLE patients, while the other studies focused on other forms of epilepsy or included a heterogeneous group of epilepsy forms. However, the authors themselves acknowledge that their sample size was rather small because of the high variability in antiepileptic drug treatments and daily drug dose; therefore, the results of their study may have marginal statistical significance. These limitations indicate the need for further studies performed on larger samples, ideally including different patients with different types of epilepsy, seizure frequency, and ASMs treatment. It is realistic to assume that no biomarker considered individually will achieve 100% prognostic or diagnostic specificity and that rather a combination of different biomarkers together with clinical information will be more likely to be used in clinical practice [53]. Finally, Tanaka et al. investigated the role of a c-miRNA as a marker for patients with acute leukemia and revealed that although miR-92a is overexpressed in lymphoma cells, it is underregulated at the plasma level. Therefore, the authors suggested that microRNAs are enclosed within exosomes secreted by cells; however, these exosomes are trapped and cannot be released, and consequently, miR-92a decreases in the blood [106]. This interesting viewpoint opens new perspectives in the interpretation of the different expressions between blood and tissues of each miRNA.

## 9. Challenges and Future Directions

Data presented in this review lead us to hypothesize that c-miRNAs are attractive candidates for the diagnosis of epileptogenesis, epilepsy, or acute seizures. It has been demonstrated that changes in miRNAs expression in body fluids occur earlier than traditional biomarkers. An exciting opportunity indeed relies on the possible use of c-miRNAs not only for earlier non-invasive diagnosis to estimate the state of epileptic disorder and the efficacy of treatments, but also to predict the risk factors for the development of SE in epileptic patients, aiding in the design of more personalized therapeutic approaches in TLE. Furthermore, c-miRNAs are of particular interest in the clinical field, as serum samples from patients with mTLE are easily accessible compared to brain tissue samples, especially in studies on childhood TLE, and are constantly circulating in the blood plasma in a stable form, being much more abundant in the plasma than in the brain. Unfortunately, there are limited studies in children with TLE compared to studies in adults. Several c-miRNAs are differentially expressed at baseline or post-seizure blood samples from patients with drug-resistant epilepsy in experimental models of TLE; however, whether some of the circulating pool of miRNAs are derived from the brain remains to be determined. To date, only a recent study has found certain c-miRNAs that originated from neurons in experimental TLE [44]. Additional studies are needed to further investigate the brain cellular origins of c-miRNAs in epilepsy models and patients. In addition, the blood plasma miRNAs may also represent an important source of diagnostic biomarkers for TLE that could also support clinical studies of antiepileptogenesis or disease-modifying therapies.

The greatest Interest in the diagnostic value of plasma miRNAs levels is their role as biomarkers of the development of therapeutic resistance to ASMs, as reported by different studies [16,28,97,98]. Nevertheless, further studies are needed to better understand the c-miRNAs correlation in the TLE mechanisms and provide safe and feasible strategies to modulate c-miRNAs for the treatment of epilepsy or prevention of epileptogenesis in TLE. Different studies on the expression level of miRNAs in biofluid have found changes in both serum and plasma at various stages of epilepsy development, showing the feasibility of detecting miRNAs in blood in both experimental models and human epilepsy. There are studies on miRNA plasma levels in preclinical and clinical TLE and it has been proposed a standardized sampling protocol for the discovery of c-miRNA biomarkers in rat models of epileptogenesis. Such a protocol could facilitate clinical biomarker discovery for human epileptogenesis [107].

In preclinical and clinical TLE studies, both plasma and serum are used for extracellular miRNA detection. Despite large progress in finding the association of specific c-miRNAs in epilepsy, factors that may affect the measurement of c-miRNAs concentrations have yet to be fully delivered.

While the miRNA profiles of plasma and serum are comparable in general terms, subtle changes in the expression patterns may occur. miRNA studies suggest that plasma may be the sample of choice in studying c-miRNAs, since miRNAs released during the coagulation process may change the true repertoire of c-miRNAs. Therefore, the plasma is less prone to certain contamination issues in which serum is affected. However, it remains to be investigated if serum or plasma is the better source of c-miRNAs biomarkers for human TLE. There is currently a lack of endogenous controls for serum or plasma miRNAs, and additional studies are necessary to compare miRNAs efficiency between serum and plasma in patients with TLE disorder. Also, the different expression of c-miRNAs among various studies can be due to the influence of the onset and course of TLE and drug resistance. There is still a lack of prospective longitudinal studies conducted on patient samples obtained at the time of diagnosis of TLE before starting ASM treatment. These ASMs can be a major confusing factor in the observed differences in c-miRNAs levels in different studies.

Overall, data reported in this review are only a small portion of the c-miRNAs studies so far, which have been associated with TLE and performed with blood samples collected from adult patients; conversely, the use of c-miRNAs as biomarkers for TLE remains scarcely investigated in epileptic children. The c-miRNAs-related mechanisms of early-life epilepsy would contribute to a rapid and precise diagnosis of TLE, as well as to the design of preventive strategies that improve the life quality of TLE children. Therefore, considering the elevated occurrence of TLE in children and adults, additional studies of the diagnostic and prognostic significance of c-miRNAs in epileptic children as potential biomarkers are foreseen.

In conclusion, c-miRNAs show strong promise as blood-based biomarkers in TLE clinical management as they are implicated in various molecular mechanisms underlying the epileptic disorder, and they can be utilized as biomarkers for better diagnosis and/or evaluation of epileptic progression. However, additional studies are needed in this area, particularly including multicenter and international clinical studies, and developing or improving the technologies for their detection in biofluids. The determination of miRNA plasma levels could indeed represent a valuable instrument to support the diagnosis and predict the prognosis of patients with TLE, in association with EEG, neuroimaging, and clinical history.

## Data Availability

Not applicable.

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
