# Peer review of "Circulating miRNAs as Novel Clinical Biomarkers in Temporal Lobe Epilepsy"

_ncrna, 2024, doi:10.3390/ncrna10020018_

Round 1

Reviewer 1 Report

Comments and Suggestions for Authors

The authors decided to conduct a review regarding the potential of circulating miRNA in the diagnosis of epilepsy (specifically in temporal lobe epilepsy, TLE), addressing issues of treatment resistance and the possible role of miRNAs as new targets in epilepsy treatment. Although extended review paper have been recently published (e.g. Mageed et al., 2024) about similar aspects of miRNAs in epilepsy, the presented review focused specifically on TLE.

First, the authors presented a bird's-eye perspective: the general aspects of TLE, provided an extended paragraph about miRNA biogenesis, and then they focused on six candidate miRNAs with biomarker potential in TLE and possible new targets for treatment. It is commendable that the authors highlighted brain-enriched miRNAs.

The review is clearly written and could be interesting for clinicians as well as pre-clinical scientists. There are only some minor aspects to consider and eventually broaden.

Regarding the treatment of TLE and the proposed six miRNAs – is there any data about the influence of anti-epileptic drugs on any of the selected miRNAs, clinically or in animal models? If yes, it could be mentioned in paragraphs regarding each proposed miRNA.

In my opinion, the paragraph "Challenges and Future Directions" is quite disappointing, especially regarding future directions, and it could be extended based on the provided data. For example, which source of miRNA (whole blood, serum, or plasma) gives the most replicable data and should be considered for future studies in this field? Are, in the opinion of the authors, the proposed six targets applicable to adult and pediatric patients alike?

Author Response

Reply to Reviewer #1:

The authors decided to conduct a review regarding the potential of circulating miRNA in the diagnosis of epilepsy (specifically in temporal lobe epilepsy, TLE), addressing issues of treatment resistance and the possible role of miRNAs as new targets in epilepsy treatment. Although extended review paper have been recently published (e.g. Mageed et al., 2024) about similar aspects of miRNAs in epilepsy, the presented review focused specifically on TLE.

First, the authors presented a bird's-eye perspective: the general aspects of TLE, provided an extended paragraph about miRNA biogenesis, and then they focused on six candidate miRNAs with biomarker potential in TLE and possible new targets for treatment. It is commendable that the authors highlighted brain-enriched miRNAs.

The review is clearly written and could be interesting for clinicians as well as pre-clinical scientists. There are only some minor aspects to consider and eventually broaden.

Response: Thank you very much for your helpful and thoughtful suggestions and comments on our manuscript.

Regarding the treatment of TLE and the proposed six miRNAs – is there any data about the influence of anti-epileptic drugs on any of the selected miRNAs, clinically or in animal models? If yes, it could be mentioned in paragraphs regarding each proposed miRNA.

Replay: We appreciate this comment; however, we have added a section entitled “Insights into the effect of miRNAs target on  antiseizure medication” in which we have discussed the available data relating to the influence of antiseizure medication on some of the selected six miRNAs.

In my opinion, the paragraph "Challenges and Future Directions" is quite disappointing, especially regarding future directions, and it could be extended based on the provided data. For example, which source of miRNA (whole blood, serum, or plasma) gives the most replicable data and should be considered for future studies in this field? Are, in the opinion of the authors, the proposed six targets applicable to adult and pediatric patients alike?

Replay: We have extended this section with additional data, and new findings that could provide important advances in this field.  We have also reported outlooks and next steps on the potential of blood-based miRNAs as biomarkers of epilepsy and epileptogenesis. In the same section, the data on the source of miRNA (serum, or plasma) has been discussed.

Regarding this request: “Are, in the opinion of the authors, the proposed six targets applicable to adult and pediatric patients alike?”

We found that most data regarding microRNA blood level changes in epileptic adult patients and there are few studies on the function of circulating microRNAs in the childhood epilepsy. These data are reported in the section of six c-miRNAs.

Reviewer 2 Report

Comments and Suggestions for Authors

In the current manuscript, the authors review existing knowledge on circulating miRNAs related to temporal lobe epilepsy. Several miRNAs have been selected and their potential as biomarkers or therapeutic targets for temporal lobe epilepsy is discussed. I have a few suggestions which I hope can help enhance the quality of the review.

  1. 1. The content in the second and third sections, in my view, does not significantly contribute to the article's theme, and I recommend reducing them. Additionally, I suggest incorporating a discussion on current diagnostic methods for TLE and their shortcomings, which could elucidate the necessity for novel biomarkers.

  2. 2. The information presented in Table 1 was not thoroughly explained in the text. I strongly recommend either modifying the content or Table 1 to ensure consistency.

  3. 3. The classification of miRNAs confused me. The authors mentioned miRNAs related to the disease's progression, drug resistance, and development. However, all these were labeled as "biomarkers" of TLE. I suggest that the authors categorize these miRNAs based on their roles in TLE and discuss their potential therapeutic roles separately. For instance, some miRNAs might be optimal as early diagnosis biomarkers, others as markers for treatment outcomes, and some for monitoring disease progression.

  4. 4. In Table 1, the regulation of certain miRNAs, such as miR-134-5p, shows varied patterns in different studies (references 62 and 67). Please include a discussion and explanation of why miRNAs behave differently and how, given these variances, they can be utilized as biomarkers.

Comments on the Quality of English Language

The use of abbreviation need to be revised. 

Abbreviation should be defined the first time it appeared in the content and abbreviation should be used in later content.

Author Response

Reply to Reviewer #2:

In the current manuscript, the authors review existing knowledge on circulating miRNAs related to temporal lobe epilepsy. Several miRNAs have been selected and their potential as biomarkers or therapeutic targets for temporal lobe epilepsy is discussed. I have a few suggestions which I hope can help enhance the quality of the review.

  1. The content in the second and third sections, in my view, does not significantly contribute to the article's theme, and I recommend reducing them. Additionally, I suggest incorporating a discussion on current diagnostic methods for TLE and their shortcomings, which could elucidate the necessity for novel biomarkers.

Replay: We have reduced the two sections (second and third) as suggested. In addition, we have included a small discussion on diagnosis methods and the possibility of using circulating miRNAs as biomarkers in TLE at the end of section 2.

  1. The information presented in Table 1 was not thoroughly explained in the text. I strongly recommend either modifying the content or Table 1 to ensure consistency.

Replay: We thank the reviewer for this important observation. We have expanded some parts in the text to better explain the data in the table and to improve the consistency of the content. In addition, we have updated the table to highlight key results on c-miRNA levels while removing  results for other biofluids that might be confounding.

  1. The classification of miRNAs confused me. The authors mentioned miRNAs related to the disease's progression, drug resistance, and development. However, all these were labeled as "biomarkers" of TLE. I suggest that the authors categorize these miRNAs based on their roles in TLE and discuss their potential therapeutic roles separately. For instance, some miRNAs might be optimal as early diagnosis biomarkers, others as markers for treatment outcomes, and some for-monitoring disease progression.

Replay: We have better described the classification of c-miRNAs following the year of paper publication and adding for each c-miRNAs the possible role as biomarkers in section 4. We have also added a paragraph that summarizes the diagnostic e/o prognostic role of different c-miRNAs in TLE (section 7).

  1. In Table 1, the regulation of certain miRNAs, such as miR-134-5p, shows varied patterns in different studies (references 62 and 67). Please include a discussion and explanation of why miRNAs behave differently and how, given these variances, they can be utilized as biomarkers.

Replay: We have added a paragraph titled “Different miRNAs behavior and their variability as biomarkers” to better explain and discuss all the probable reasons why miRNAs might give discordant results (section 8).

Finally, by mutual agreement, we have included among the authors Dr. Stefano Ruga, who contributed to the revision phase of the manuscript.

Round 2

Reviewer 2 Report

Comments and Suggestions for Authors

I appreciate the authors' response. All my points have been well addressed. There is still some very minor suggestion about the revised manuscript.

1. abbreviation use.

If the full name only appeared once in the whole content, no abbreviation is needed (example, DGCR8, BBB). The abbreviation should be defined the first time a full name is given. For example, abbreviation "CNS" first appeared at line 89, but full name appeared at line 169. After an abbreviation is defined, the full name should not be used again (example, after-seizure sample (EAS) was defined at line 358, but full name was used in table 1).

2. In table 1, I believe the method should be "qRT-PCR" (as mentioned in the main content) instead of "RT-PCR".

3. Possible typo "RQ-PCR" in table 1.

4. Formatting issues. In table 1, some if the "vs" were italic while some are not. Please keep the format consistent.

I would suggest the authors go through the manuscript and correct these issues. 

Author Response

Responses to Reviewer 2

  1. abbreviation use.

If the full name only appeared once in the whole content, no abbreviation is needed (example, DGCR8, BBB). The abbreviation should be defined the first time a full name is given. For example, abbreviation "CNS" first appeared at line 89, but full name appeared at line 169. After an abbreviation is defined, the full name should not be used again (example, after-seizure sample (EAS) was defined at line 358, but full name was used in table 1).

We thank the reviewer for the suggestion. We have made the required changes to the abbreviations. For the ''blood-brain barrier (BBB)'' (reported in two places: on lines 167 and 663) and for "central nervous system (CNS)" (reported in two places: on lines 89 and 169) we left the abbreviation. Instead, we have eliminated some acronyms such as FCD, ILAE, PET etc. because, as suggested, the term was reported only once in the entire text.

  1. In Table 1, I believe the method should be "qRT-PCR" (as mentioned in the main content) instead of "RT-PCR".

We thank the reviewer for the suggestion. We have made the requested change.

  1. Possible typo "RQ-PCR" in table 1.

We thank the reviewer for this observation. We have corrected the beat error in the table.

  1. Formatting issues. In table 1, some if the "vs" were italic while some are not. Please keep the format consistent.

We thank the reviewer for this suggestion, and we have standardized ‘’vs’’ in the table.